# Chimeric Measles Virus (MV/RSV), Having Ectodomains of Respiratory Syncytial Virus (RSV) F and G Proteins Instead of Measles Envelope Proteins, Induced Protective Antibodies against RSV

**DOI:** 10.3390/vaccines9020156

**Published:** 2021-02-16

**Authors:** Akihito Sawada, Takashi Ito, Yoshiaki Yamaji, Tetsuo Nakayama

**Affiliations:** Laboratory of Viral Infection I, Ōmura Satoshi Memorial Institute, Kitasato University, Tokyo 108-8641, Japan; akihito@lisci.kitasato-u.ac.jp (A.S.); itot@lisci.kitasato-u.ac.jp (T.I.); di12004@st.kitasato-u.ac.jp (Y.Y.)

**Keywords:** measles AIK-C, respiratory syncytial virus (RSV), ectodomain, recombinant chimeric virus

## Abstract

In our previous study, fusion (F) or glyco (G) protein coding sequence of respiratory syncytial virus (RSV) was inserted at the P/M junction of the measles AIK-C vector (MVAIK), and the recombinant measles virus induced protective immune responses. In the present study, the ectodomains of measles fusion (F) and hemagglutinin (HA) proteins were replaced with those of RSV F and G proteins, and a chimeric MV/RSV vaccine was developed. It expressed F and G proteins of RSV and induced cytopathic effect (CPE) in epithelial cell lines (Vero, A549, and HEp-2 cells), but not in lymphoid cell lines (B95a, Jurkat, and U937 cells). A chimeric MV/RSV grew similarly to AIK-C with no virus growth at 39 °C. It induced NT antibodies against RSV in cotton rats three weeks after immunization through intramuscular route and enhanced response was observed after the second dose at eight weeks. After the RSV challenge with 10^6^ PFU, significantly lower virus (10^1.4±0.1^ PFU of RSV) was recovered from lung tissue in the chimeric MV/RSV vaccine group than in the MVAIK control group with 10^4.6±0.2^ PFU (*p* < 0.001) and no obvious inflammatory pathological finding was noted. The strategy of ectodomain replacement in the measles virus vector is expected to lead to the development of safe and effective vaccines for other enveloped viruses.

## 1. Introduction

Respiratory syncytial virus (RSV) is a negative sense single-stranded RNA virus belonging to the subfamily Pneumovirinae. RSV is a common infectious agent causing respiratory illness [1,2,3]. It causes serious lower respiratory infections (pneumonia, bronchiolitis, and pneumonitis) among young infants with congenital heart diseases, low birth weight, congenital abnormality, Down’s syndrome, and immunodeficiency. Approximately 70% of children are infected with RSV by one year of age and almost all by two years of age [4]. Repeated infections are observed because of poor immune responses in the younger generation; respiratory symptoms are mitigated in older children. Since RSV is such a common pathogen, the WHO estimated 33 million patients with serious RSV infection per year, 3 million hospitalizations, and 60,000 deaths in children <5 years globally [5].

Humanized monoclonal antibody medicines are used prophylactically, but their clinical usage is limited for high-risk children. They should be administered monthly, which is an enormous medical financial burden [6]. Therefore, preventive vaccines are expected. Although many pipelines for vaccine development have been investigated, there is no vaccine currently available for clinical use. In the 1960s, formalin-inactivated RSV (FI-RSV) vaccines were administered to young infants, but serious lower respiratory illness developed after natural infection among 31 FI-RSV recipients. Among them, 80% were hospitalized and two died. Autopsy findings showed marked infiltration of mononuclear cells, eosinophils, and polymorphonuclear cells. Extensive research on animal models has been reported along with explanatory theories: it induced only binding antibodies in serum without neutralizing activity, there was no secretory IgA response, and there were skewed Th2-dominant immune responses with allergic reactions. These should be considered in the development of an RSV vaccine [7,8,9,10].

Several approaches have been adopted to develop RSV vaccines [11,12,13]. Conventional serial passage in mammalian cells at a lower temperature did not establish a live attenuated vaccine candidate. More recently, attenuating mutations have been inserted into vaccine candidates using reverse genetics, but suitable candidates have not yet been developed [14]. RSV consists of two envelope proteins, fusion (F) and glyco (G) proteins, considered targets for neutralization, and nucleo (N) protein for cytotoxic T lymphocytes (CTL) [11,12,13]. Chimeric bovine/human parainfluenza virus expressing RSV F protein progressed to a phase I trial, but was suspended because of poor immune responses against RSV, and the insertion position of the RSV F gene was reconsidered [15,16]. We developed the measles vaccine AIK-C strain that expressed the F (MVAIK/RS/F), G (MVAIK/RS/G), and N proteins (MVAIK/RS/N). A higher and prolonged neutralization test (NT) antibody was observed in cotton rats immunized with MVAIK/RS/F and higher levels of CTL in those immunized with MVAIK/RS/N [17,18,19]. Cotton rats immunized with these recombinant vaccines were protected from RSV challenge. They were constructed by the insertion of coding regions for each protein at the P/M junction of the infectious cDNA of the AIK-C vaccine strain. Using this construction strategy, we developed recombinant measles vaccines expressing prM+E protein of Japanese encephalitis [20] and influenza hemagglutinin (HA) protein, and they induced protective immune responses [21].

Measles virus is classified in the subfamily Paramyxovirinae and has two functional envelope proteins, F and HA. The RSV and measles viruses belong to the same family, Paramyxoviridae, and have similar genome structures to those of F and G/HA genes. Conventional approaches failed to develop RSV vaccine candidates that had the temperature sensitivity (*ts*) phenotype. The AIK-C measles vaccine has unique *ts* characteristics contributing to attenuation mechanisms and MVAIK constructed by reverse genetics also has the *ts* phenotype [22]. MVAIK/RS/F or MVAIK/RS/G has measles envelope proteins of F and HA and their efficacy may be influenced by pre-existing antibodies. In the present study, the chimeric RSV vaccine candidate was constructed based on the AIK-C measles virus, replacing the ectodomains of the F and HA proteins of the measles virus with those of the F and G proteins of RSV. The chimeric measles and RSV vaccine (MV/RSV) was characterized and we investigated the immune response.

## 2. Materials and Methods

### 2.1. Virus Strains and Cells

The AIK-C strain for vaccine seed was used for reverse genetics. Wild-type strain of RSV subgroup A isolated in 2004 was propagated in HEp-2 cells from the patient. Long strain was used for the neutralization test (NT) against RSV. 293T, A549, and HEp-2 cells were maintained in Eagle’s MEM (Sigma–Aldrich, Dorset, UK) supplemented with 10% fetal bovine serum (FBS). Vero cells were maintained in Eagle’s MEM supplemented with 5% FBS. B95a, Jurkat, and U937 cells were maintained in RPMI-1640 medium (Sigma-Aldrich, Dorset, UK) supplemented with 10% FBS. These culture media were supplemented with 4 mM L-glutamine, 10,000 IU/mL penicillin, and 10,000 μg/mL streptomycin. All cells were cultured at 37 °C in 5% CO_2_. Vero, 293T, and B95a cells were provided by Kitasato Daiichi Sankyo Vaccine, and the other cells were purchased from ATCC.

### 2.2. Construction of Recombinant Chimeric AIK-C (MV/RSV)

Backbone of MVAIK was constructed from the AIK-C vaccine strain developed in our laboratory, which has been used as the recommended form of immunization in Japan since 1977 as it has the *ts* phenotype [22]. A schematic diagram of the strategy used for the construction of MV/RSV cDNA plasmid is shown in Figure 1. The expression plasmids of RSV F and G proteins were constructed in our previous study [17]. To replace the ectodomains of measles F and HA proteins with those of RSV F and G proteins, expression plasmids of the chimeric MV/RSV/F and MV/RSV/G proteins were constructed using transition primer sets. Transition primers were designed to switch at the border between transmembrane (TM) and ectodomain regions considering the sequential hydrophobic amino acids. TM and cytoplasmic (CT) regions are maintained and the construction strategy is shown in detail in Appendix A. The chimeric MV/RSV/G protein sequence was introduced into the part of MVAIK cDNA at *Sal* I (genome position 3364) and *Spe* I (genome position 9175). The chimeric MV/RSV/F protein sequence was inserted at *Nar* I (genome position 4922) and *Pac* I (genome position 7238). Full-length infectious chimeric cDNA, with RSV F and G ectodomains with measles TM and CT, was constructed based on an MVAIK cDNA clone.

Next, 293T cells were infected with MVA T7 Pol expressing T7 RNA polymerase and pMV/RSV was transfected together with helper plasmids encoding the N, phospho (P), and large (L) proteins of the AIK-C using Trans IT-LT1 Reagent (Mirus Bio Corporation, Madison, WI, USA). The medium was replaced with fresh MEM supplied with 5% FBS after incubation for 3 h. The 293T cells were detached after a 2-day culture and co-cultured with Vero cells [17,18,19,20,21].

### 2.3. Virus Culture and Purification

To examine the viral growth at different temperatures, Vero cells were infected with MVAIK, MV/RSV, and RSV (m.o.i. = 0.1) and the plates were placed at different temperatures of 33, 35, 37, and 39 °C in 5% CO_2_. The culture fluids were obtained on days 1, 3, 5, and 7 of culture, and infective titers were examined and expressed as TCID50/mL in Vero cells.

Vero cells were infected with MVAIK and HEp-2 were infected with MV/RSV and RSV wild-type isolate. Culture fluid was collected and fractionated through sucrose discontinuous gradient ultra-centrifugation of 30%, 45%, and 60%. Purified virus particle fraction was obtained between 45% and 60% sucrose.

### 2.4. Immunostaining

Vero cells were infected with MVAIK, MV/RSV, and RSV at m.o.i. of 0.1 in 24-well plates and cultured for three days at 33 °C. Vero cells were fixed with 1% glutaraldehyde for 30 min and subjected to indirect immunostaining. Infected cells were incubated with monoclonal antibodies against RSV G (HyTest, Turku, Finland) for 1 h at 37 °C. The cells were washed extensively with PBS (-) with 0.05% Tween 20 (PBST) and stained with a secondary antibody against anti-mouse IgG conjugated with FITC (Proteinteck, Rosemont, IL, USA). As for the detection of the RSV F protein, a monoclonal antibody against RSV F protein (Abcam, Cambridge, UK) and an anti-mouse IgG conjugated with FITC (Proteinteck, IL, USA) were used. Vero cells were also incubated with a mouse monoclonal antibody against MV HA protein (kindly supplied by Dr. Sato, National Institute of Infectious Diseases, Tokyo, Japan) and followed by a secondary antibody against mouse IgG conjugated with rhodamine raised in goat (Rockland Immunochemicals, Gilbertsville, PA, USA). The expression of measles N protein was stained with a monoclonal antibody against measles N protein (Abcam, Cambridge, UK) and a secondary antibody conjugated with Alexa Fluor 568 (Invitrogen, Carlsbad, CA, USA).

HEp-2 cells were infected with purified MV/RSV, RSV wild-type, and MVAIK and stained using a goat polyclonal antibody against RSV (Abcam Cambridge, England, UK), an anti-goat IgG antibody conjugated with horseradish peroxidase (Santa Cruz Biotechnology, Inc., Dallas, TX, USA), and a DAB Stain (Nacalai Tesque, Inc., Kyoto, Japan). Microscopic images were taken using EVOS FL light microscope (Life technologies, Waltham, MA, USA).

### 2.5. Viral Tropism

To investigate the cell tropism, Vero, A549, and HEp-2 cells were infected with MVAIK, MV/RSV, and RSV (m.o.i. = 0.1). B95a, Jurkat, and U937 cells were also infected in a similar way. Microscopic images to assess the appearance of the cytopathic effect (CPE) were taken using a Life Technologies EVOS XL Core light microscope (Life technologies, USA).

### 2.6. Immunization and RSV Challenge in Cotton Rats

Six-week-old inbred female cotton rats (*Sigmodon hispidus*) were used. Three cotton rats for each group were immunized with 1 × 10^5^ TCID_50_ of MVAIK and MV/RSV intramuscularly (i.m.) or intranasally (i.n.). Serum samples were obtained immediately before and 1, 3, 5, 8, 9, 12, and 15 weeks after immunization. The immunization experiment was performed in duplicate. They were re-immunized at 8 weeks after the first dose and were challenged with RSV four weeks after reimmunization (12 weeks after the first dose). Briefly, cotton rats were anesthetized and challenged with 10^6^ PFU/0.5 mL of RSV/Long through intranasal administration. They were sacrificed four days after the challenge and lung tissues were obtained. Lung samples were divided into two portions, one for pathological examination and another for recovering the infectious viral particles. Experimental protocol was approved by the Committee of Experimental Animal Study of the Kitasato Institute for Life Sciences (No. 16-026, 17-027, and 18-030).

Tissues were homogenized and 0.1 mL volume of serial 10-fold dilutions of homogenized samples were placed on HEp-2 cells and were then overlaid with MEM 5% FBS and 0.5% agar. Plaque numbers were counted after incubation for six days at 37 °C and infectivity was expressed as the number of plaques adjusted to 50 mg of lung tissue.

Lungs were inflated with 4% formalin and submerged in formalin for overnight fixation. The fixed tissue was embedded in paraffin, sectioned, and stained with hematoxylin–eosin. Microscopic figures were taken at more than 6 different sites in each group and were evaluated in a blind manner.

### 2.7. Serology

Neutralization tests against RSV were performed with a 50% plaque reduction assay using Long strain as previously reported. In brief, serum samples were serially diluted 4-fold starting from 1:10 dilution and mixed with an equal volume of approximately 100 PFU of challenge virus for 1 h at 37 °C. Serial mixtures were placed on the monolayer of HEp-2 cells in a 24-well plate, and the cells were overlaid with MEM containing 0.5% agar. After 7-day incubation period, the cells were fixed with glutaraldehyde and the agar was removed. Plaque numbers were counted after staining with neutral red and NT antibody titers were calculated as the reciprocal of the serum dilutions that showed a 50% reduction in plaque numbers as previously reported [17,18].

Particle agglutination (PA) titers were assayed using a detection kit (Serodia^®^-Measles, Fuji Rebio, Tokyo, Japan). Serial 2-fold dilutions were mixed with gelatin particles coated with measles antigen starting at 1:10 dilution. PA titers were defined as the reciprocal of the highest dilutions of agglutination [23].

### 2.8. Statistical Analysis

Statistical analysis was performed by Welch’s *t*-test for infectious virus recovery after an RSV challenge test.

## 3. Results

### 3.1. Virus Growth

Vero cells were infected with MVAIK (empty vector), chimeric MV/RSV, and RSV/Long. Similar growth rates were obtained for the three viruses with peak infectivity on day 5 after infection (Figure 2A). Virus growth was compared at different temperatures, and culture fluids were obtained on day 5 of infection. Infective titers are shown in Figure 2B. Highest infectivity of RSV/Long was observed at 33 °C and approximately similar 10^4-5^ TCID 50 was observed at 35 °C and 37 °C. Although RSV/Long showed similar virus infectivity at 39 °C, no virus growth was noted for MVAIK and chimeric MV/RSV at 39 °C, demonstrating original temperature sensitivity (*ts* phenotype).

### 3.2. Expression of Envelope Proteins

The expression of envelope proteins was examined in Vero cells infected with MVAIK, MV/RSV, and RSV/Long. The results of immunostaining are shown in Figure 3. Measles N and HA proteins were expressed in Vero cells infected with MVAIK, but not in those infected with RSV/Long. MV/RSV was constructed using an MVAIK vector whose ectodomains of the F and HA of measles were replaced with those of the RSV F and G protein. Vero cells infected with MV/RSV showed the expression of RSV F, G, and measles N proteins, but not measles HA. The appropriate expression of the exchanged envelope proteins was confirmed, but cell fusion of chimeric MV/RSV is not demonstrable in Vero cells.

MV/RSV, RSV wild-type, and MVAIK were purified through sucrose ultra-centrifugation. Western blotting of purified MVAIK, MV/RSV, and RSV was examined, and staining with monoclonal antibody was performed against RSV F protein. RSV F protein band was detected in the lanes of electrophoresis of purified RSV and MV/RSV, but not in that of MVAIK (Figure 4A and Appendix A). Purified MV/RSV induced CPE in HEp-2 cells similar to RSV infection. RSV protein was expressed in HEp-2 cells infected with purified MV/RSV and RSV, but not MVAIK (Figure 4B).

### 3.3. Different Cellular Tropism

The measles vaccine virus infects both epithelial and lymphoid cells, but RSV only infects respiratory epithelial cells. Vero is a monkey kidney cell line, A549 is derived from alveolar cells, and HEp-2 cells come from head and neck cancer cells. They were infected with MVAIK, MV/RSV, and RSV/Long and CPE were observed in these cells (Figure 5). B95a, Jurkat, and U937 cells, representative of lymphoid cells, were infected, but only MVAIK induced CPE in these cell lines. It is noted that MV/RSV had the same cell tropism as RSV.

### 3.4. Immunogenicity of MV/RSV

Cotton rats were immunized with MVAIK and MV/RSV and reimmunized at 8 weeks after the first dose. Serial serum samples were obtained until 15 weeks after the first dose and the results of the serum antibody against measles and RSV are shown in Figure 6. Measles PA antibodies were only detected in cotton rats immunized with MVAIK and no detectable measles antibody was noted in those immunized with MV/RSV through i.m. and i.n. routes.

The NT antibody against RSV/Long strain is shown in Figure 6. Detectable NT antibody against RSV was developed 3 weeks after the first dose and the reimmunization enhanced the production of NT antibodies in cotton rats immunized with MV/RSV through i.m. administration, but not through the i.n. route.

### 3.5. Protective Effect of MV/RSV

Cotton rats were immunized with MV/RSV through i.m. and i.n. routes and MVAIK through i.m. They were challenged with 10^6^ PFU of RSV 4 weeks after reimmunization. Lung tissues were obtained four days after the challenge. Recovery of infectious RSV was examined and the results are shown in Figure 7. Likewise, 10^4.3±0.2^ PFU of RSV was recovered from 50 mg of lung tissues obtained from cotton rats immunized with MV/RSV through the i.n. route, and similar results were obtained after immunization with MVAIK and non-immunized rats. Extremely low levels of 10^1.4±0.1^ PFU of RSV were detected from lung tissue obtained from those immunized with MV/RSV through the i.m. route (*p* < 0.001).

Histopathological findings are shown in the lower panel of Figure 7. No significant pathological finding was noted in cotton rats immunized with chimeric MV/RSV through the i.m. route, like the normal cotton rat. However, destruction of alveolar structures and inflammatory responses were demonstrated in those immunized with MV/RSV through the i.n. route, MVAIK, and non-immunized control: infiltration of inflammatory cells in the peri-bronchus, thickening of the alveolar wall, and swelling of bronchial epithelial cells.

## 4. Discussion

Live vaccines have several merits to prevent infection and serious illness, but no live vaccine against RSV has been developed [11,12,13]. Therefore, several recombinant virus vaccines have been investigated using vaccinia, adeno, sendai, influenza, and measles viruses [24,25,26,27,28]. In our laboratory, recombinant measles vaccines based on an infectious cDNA clone of AIK-C were developed, expressing RSV F, G, and N proteins, peM+E of Japanese encephalitis virus, and HA protein of influenza virus [17,18,19,20,21,29]. Heterologous envelope protein coding sequences were inserted at the P/M junction. They induced protective immune responses with the development of a neutralizing antibody and CTL.

Immune responses were influenced by pre-immunization antibody levels in a clinical study using the AIK-C vaccine [30]. Low levels of measles antibody did not influence the immunogenicity of measles vaccines, but higher levels of measles antibody reduced the virus growth, resulting in poor immune responses. The pre-immunization measles immune status influenced the efficacy of recombinant vaccine candidates. Especially for RSV, live recombinant vaccine candidates should be administered to young infants before measles vaccination. Serious RSV infection occurs at <6 months and live RSV vaccine candidates should be administered around this time [13]. Vaccine efficacy of measles virus-vectored vaccine candidates may vary depending on maternal-conferred immunity or vaccine-acquired immunity at >1 year of age. Reisinger et al. [31] reported that the immunogenicity of measles-vectored chikungunya virus vaccine MV-CHIK was not influenced by pre-existing immunity, but the immunization cohort consisted of adults 18–55 years old with relatively low antibodies. Low levels of pre-immunization antibodies did not impede the development of immune response.

In the present study, the chimeric MV/RSV virus was developed and replaced the ectodomains of F and HA proteins of measles with those of RSV F and G proteins and showed the same cellular tropism as RSV. Virus growth was also investigated and MV/RSV recombinant virus failed to grow at 39 °C, which likened the *ts* phenotype to that of AIK-C. AIK-C has a unique characteristic of the *ts* phenotype; Pro at position 439 of the phospho (P) protein is responsible for the *ts* phenotype [22]. MVAIK constructed by reverse genetics from the AIK-C vaccine seed did not grow at 38 °C, indicating the stricter *ts* phenotype [22]. The body temperature of cotton rats is approximately 38 °C. In our previous study using cotton rats, no virus growth was observed when they were immunized with MVAIK and AIK-C vaccines through the i.n. route [32]. Through the i.m. route, the measles genome or infectious virus was detected in the thymus and regional lymph nodes in cotton rats [32]. RSV primarily infects the upper respiratory tract and immunization through the i.n. route is a more practically effective procedure. No detectable NT antibody against RSV was induced through the i.n. route because of their *ts* phenotype in the cotton rat model with high body temperature.

Tissue or cellular tropism is defined by viral envelope proteins and virus growth depends on RNA polymerase activity of N, P, and L proteins. Virus particle formation of the measles virus depends on the M protein and its interaction with the CT of HA and F envelope proteins. The transmembrane (TM) region of the envelope proteins of F and HA acts as an anchor to trap them at the cell membrane [33,34]. In the construction strategy, measles N, P, M, and L proteins are the backbone of AIK-C vaccine strain and TM and CT regions are retained. The ectodomains of RSV F and G were fused to respective TM and CT regions of F and HA proteins, acting as the chimeric virus particles. RSV F protein fused to the measles TM–CT domain and was present on the cell membrane of purified MV/RSV particles, which induced CPE in HEp-2 cells as well as RSV. It induced protective NT antibodies.

Comparative clinical trials showed that the standard potency of the AIK-C strain vaccine induced a stronger serological response than the other high potency measles vaccines and was expected to be used for infants <9 months in developing countries [30,35,36,37]. These biological characteristics are clinically beneficial for the development of an AIK-C vector-based vaccine for young infants.

## 5. Conclusions

In our present study, ectodomains of measles F and HA proteins were appropriately replaced with RSV F and G proteins, which were expressed on the surface of chimeric MV/RSV particles and infected cells. This result showed the same cellular tropism as RSV and the same *ts* phenotype as the parental AIK-C measles vaccine. The chimeric MV/RSV induced protective immune responses. This construction strategy of exchanging ectodomains with a heterologous virus envelope may be applicable for the development of enveloped virus vaccines such as RSV, mumps, Nipah, human metapneumovirus in the same member of family Paramyxoviridae, and COVID-19, which is now of noticeable concern regarding public health.

## Figures and Tables

**Figure 1 vaccines-09-00156-f001:**
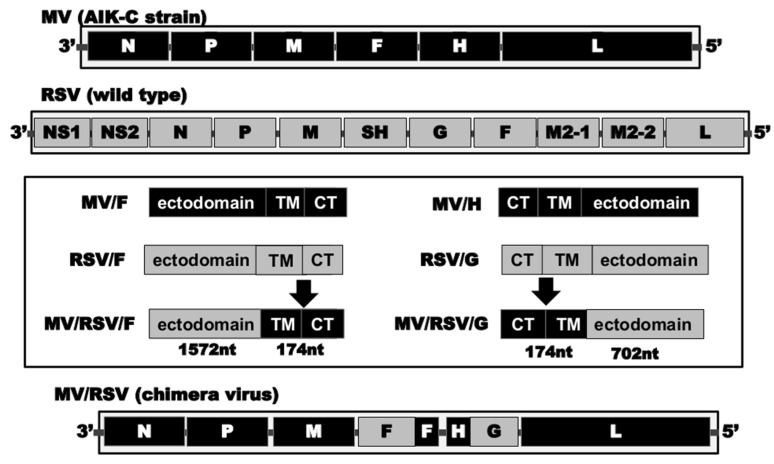
Construction strategy. Gene of the measles virus is shown in black and that of RSV in gray. The ectodomains of measles F and HA protein genomes were replaced with those of RSV F and G protein genomes. They were fused to the respective TM–CT region of the measles virus. The chimeric infectious cDNA (MV/RSV) was constructed and details are in Appendix A.

**Figure 2 vaccines-09-00156-f002:**
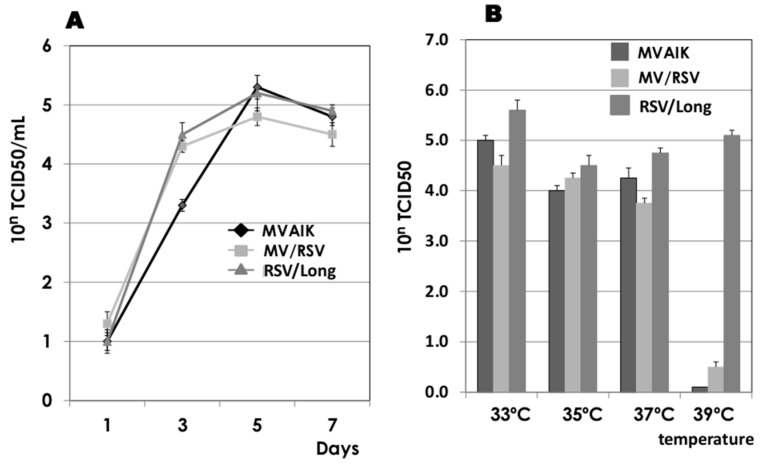
Virus growth of MVAIK, MV/RSV, and RSV/Long and comparison of virus growth at different temperatures. (**A**) Vero cells were infected with MVAIK, MV/RSV, and RSV/Long and culture fluids were harvested at 1, 3, 5, and 7 days after infection. Vertical lines show the mean ± error bar of three experiments. (**B**) Virus infectivity on day 5 of infection at different temperatures of 33, 35, 37, and 39 °C. Vertical columns show the infective virus titer of mean + error bar of three experiments.

**Figure 3 vaccines-09-00156-f003:**
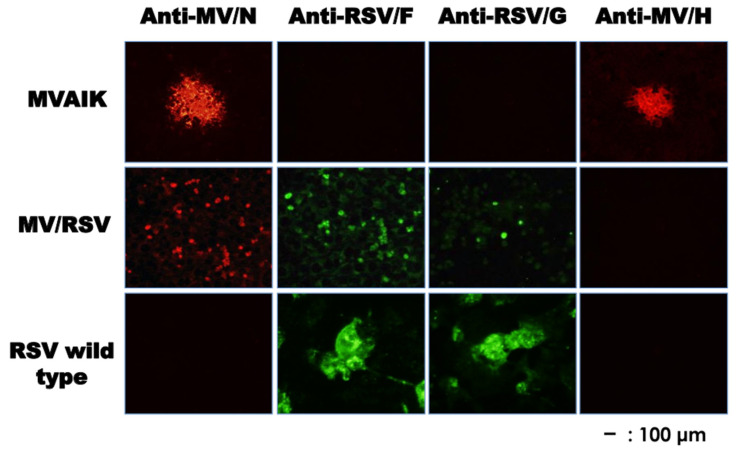
Immunostaining of Vero cells infected with MVAIK, MV/RSV, and RSV. Monoclonal antibodies against RSV F or G protein and secondary antibody against mouse IgG conjugated with FITC were used for the detection of RSV F or G protein. Monoclonal antibody against MV HA protein and secondary antibody against mouse IgG conjugated with rhodamine were used for the detection of MV HA. The expression of measles N protein was stained with monoclonal antibody against measles N protein and secondary antibody conjugated with Alexa Fluor 568.

**Figure 4 vaccines-09-00156-f004:**
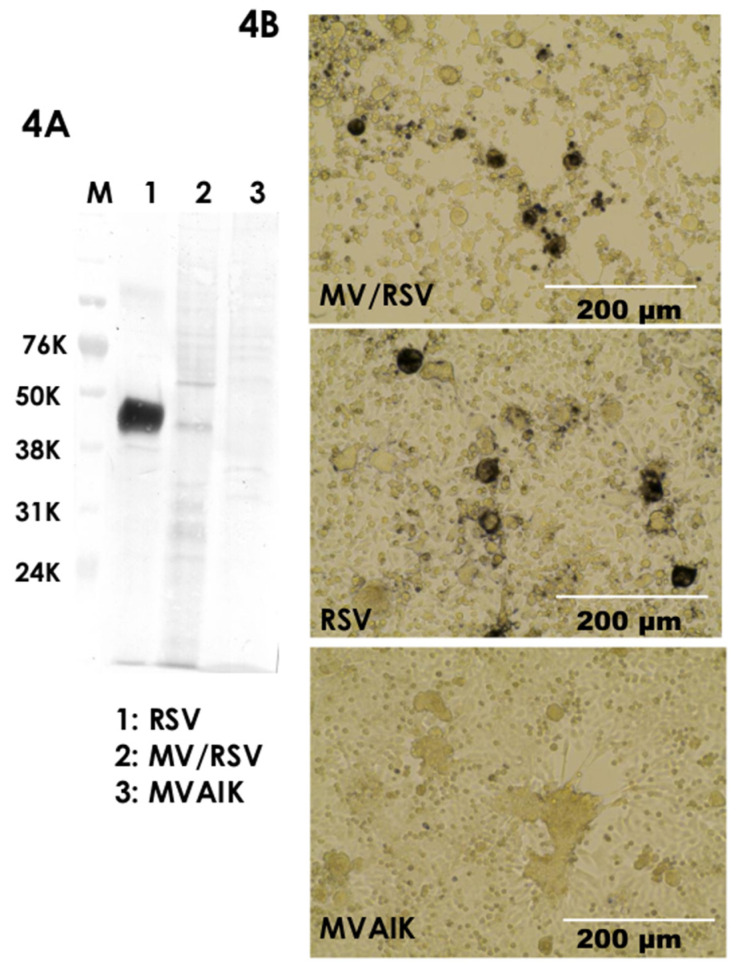
Expression of RSV F protein on MV/RSV chimera virus particle. Western blotting of the RSV, MV/RSV, and MVAIK was examined using monoclonal antibody against RSV F protein (**A**). HEp-2 cells were infected with RSV, purified MV/RSV, and MVAIK and fixed 2 days after infection using polyclonal antibodies against RSV and HRT conjugated secondary antibody against goat IgG. Microscopic image was taken ×20 by Life Technologies EVOS. White bars represent 200 μm (**B**).

**Figure 5 vaccines-09-00156-f005:**
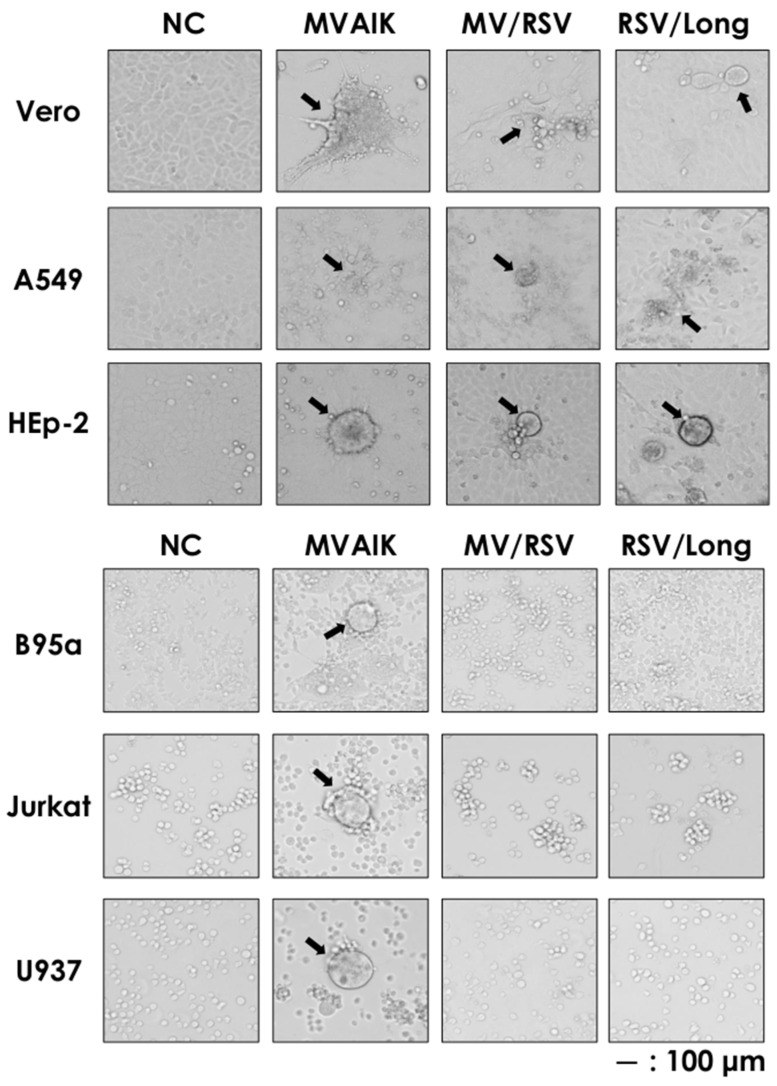
Appearance of CPEs in Vero, A 549, HEp-2, B95a, Jurkat, and U937 cells infected with MVAIK, MV/RSV, and RSV/Long. Vero, A 549, HEp-2, B95a, Jurkat, and U937 cells were infected with MVAIK, MV/RSV, and RSV. The cells were fixed and CPE was observed. Arrows indicate the appearance of cell fusion. Scale bar of 100 μm is shown.

**Figure 6 vaccines-09-00156-f006:**
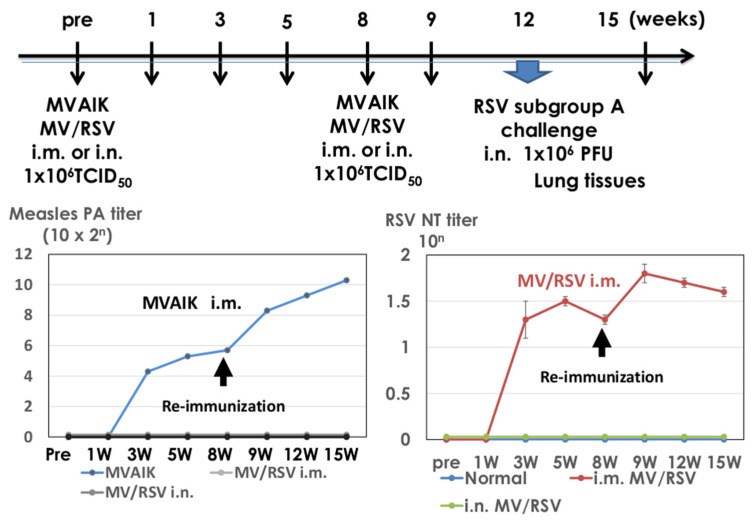
Immunization schedule and antibody responses against measles and RSV. Three cotton rats were immunized with MVAIK and MV/RSV through intra-nasal (i.n.) and intra-muscular (i.m.) routes and were reimmunized at 8 weeks after the first dose. They were challenged with 1 × 10^6^ PFU of the RSV/Long strain 4 weeks after reimmunization. Lung tissues were obtained 4 days after the challenge. Measles PA antibody and RSV NT titers were examined. Immunization experiment was performed in duplicate.

**Figure 7 vaccines-09-00156-f007:**
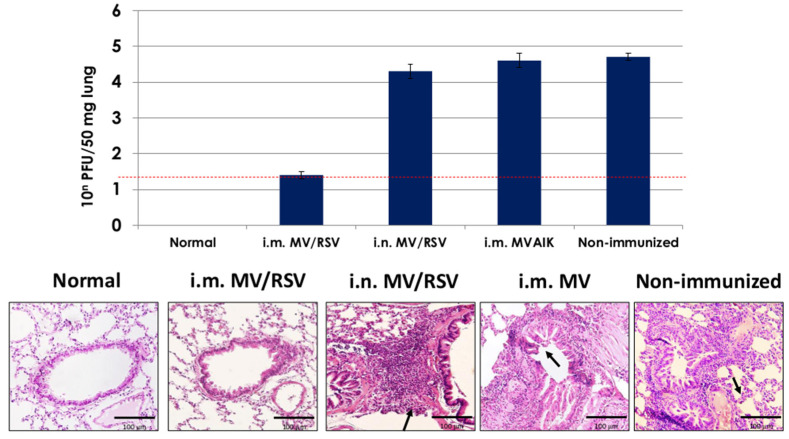
Recovery of the infectious virus in the lung tissue homogenates and the pathological findings of lung tissues. Infectivity is expressed per 50 mg lung tissue with error bars. Formalin-embedded lung tissues were sectioned and stained with HE. Arrows show the pathological findings and scale bars represent 100 μm. The red dotted line indicates detection limit of recovery of infectious RSV.

## Data Availability

Data available in a publicly accessible repository.

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
