# Peer review of "Chimeric Measles Virus (MV/RSV), Having Ectodomains of Respiratory Syncytial Virus (RSV) F and G Proteins Instead of Measles Envelope Proteins, Induced Protective Antibodies against RSV"

_vaccines, 2021, doi:10.3390/vaccines9020156_

Round 1

Reviewer 1 Report

The revised version of the manuscript improved significantly. Therefore. It is acceptable in its current form. 

Author Response

Thank you for your reviewing.

Reviewer 2 Report

This revised manuscript improved partially following reviewers' suggestions and I have some minor points for authors' information.

1) Fig.3 is invalid without a "bright field". 

2) As for the scale bar issue, if the authors use the EVOS FL platform, please find the following instructions to add a scale bar to related images

"The Scalebar is a toggle button that displays or hides the Scale bar tool. This option is only available after an image is captured. To move the Scale bar, simply click and drag it."

Author Response

Thank you for your commnets.

As for Figure 3. I tried to change to more blight fields, but the expression of each protein is clear without change the blightness.

Scale bars were included in Figures 3, 4, 5, and 7.

This manuscript is a resubmission of an earlier submission. The following is a list of the peer review reports and author responses from that submission.

Round 1

Reviewer 1 Report

Sawada et al., presents a study where they have analyzed the replacement of the ectodomains of F and HA proteins of AIK-C measles vaccine vector with the respiratory syncytial virus F and G proteins to generate chimeric MV/RSV vaccine. Moreover, authors also investigate the immune response of this chimeric MV/RSV vaccine using cotton rat model. This is an interesting study, but the current form of this article does not meet the journal standard. The major drawback of this study is no statistical analysis were performed. In addition, authors did not mention that how many times the experiments were conducted. Therefore, it is extremely hard to evaluate the presented data in the manuscript. Authors should be more careful about the study design and presentation. There are quite a few questions/concerns about this study, however I have summarized few of them here-

  1. Abstract needs to rewritten completely. As a way it is presented here does not clearly reflect the significance of the study (i.e. why they performed the current study and what they have done new.)
  2. What is the source of all cell lines were used in this study?
  3. Authors did not mention if the animal experiments of the study were performed in compliance with all federal and local guidelines and approved by the Institutional Animal Committee. Moreover, sex of the animal needs to include in the methods section.
  4. Regarding the Histopathology I wanted to know that if the evaluation was performed from blinded slides or not. If so, please include that information in the manuscript.
  5. In Figure 4, the live staining of RSV long and MV/RSV infected HEp-2 cells to determine the expression of F protein is not clear. The representative picture has lot of backgrounds making it extremely difficult to see the expression of F protein in cells. A much clearer picture/data is expected.

Author Response

Response to Reviewer 1 Thank you for your comment.

【Comment】The major drawback of this study is no statistical analysis were performed. In addition, authors did not mention that how many times the experiments were conducted. Therefore, it is extremely hard to evaluate the presented data in the manuscript. Authors should be more careful about the study design and presentation.

【Response】 Three cotton rats were used in immunization experiments. Immunization experiment was performed twice, and similar results were obtained. There was no need to analyze the statistical differences in this study.

【Comment】Abstract needs to rewritten completely. As a way it is presented here does not clearly reflect the significance of the study (i.e. why they performed the current study and what they have done new.)

【Response】 Some parts of the previous study were deleted and abstract was rewritten as following;

In our previous study, fusion (F) or glyco (G) protein coding sequence of respiratory syncytial virus (RSV) was inserted at the P/M junction of the measles AIK-C vector (MVAIK), and the recombinant measles virus induced protective immune responses. In the present study, the ectodomains of measles fusion (F) and hemagglutinin (HA) proteins were replaced with those of RSV F and G proteins, and a chimeric MV/RSV vaccine was developed. It expressed F and G proteins of RSV and induced cytopathic effect (CPE) in epithelial cell lines (Vero, A549, and HEp-2 cells) but not in lymphoid cell lines (B95a, Jurkat, and U937 cells). A chimeric MV/RSV grew similarly to AIK-C, with no virus growth at 39ºC. It induced NT antibodies against RSV in cotton rats 3 weeks after immunization through intramuscular route, and enhanced response was observed after the second dose at 8 weeks after the first dose. After the RSV challenge with 106 PFU, significantly lower virus (101.4±0.1 PFU of RSV) was recovered from lung tissue in the chimeric MV/RSV vaccine group than in the MVAIK control group with 104.6±0.2 PFU, and no obvious inflammatory pathological finding was noted. The strategy of ectodomain replacement in measles virus vector is expected to lead to the development of safe and effective vaccines for other enveloped viruses.

【Comment】 What is the source of all cell lines were used in this study?

【Response】Following sentence was added, line 100-101: Vero, 293T, and B95a cells were provided by Kitasato Daiichi Sankyo Vaccine, and the other cells were purchased from ATCC.

【Comment】Authors did not mention if the animal experiments of the study were performed in compliance with all federal and local guidelines and approved by the Institutional Animal Committee. Moreover, sex of the animal needs to include in the methods section.

【Response】All animals were female. It was added, line 165. As for the guideline, following sentence was added, line 173-175: Experimental protocol was approved by the Committee of Experimental Animal Study of Kitasato Institute for Life Sciences (No. 16-026, 17-027, and 18-030).

【Comment】Regarding the Histopathology I wanted to know that if the evaluation was performed from blinded slides or not. If so, please include that information in the manuscript.

【Response】 Following sentence was added, line 182-183: Microscopic figures were taken at more than 6 different sites in each group and were evaluated in a blind manner by us.

【Comment】In Figure 4, the live staining of RSV long and MV/RSV infected HEp-2 cells to determine the expression of F protein is not clear. The representative picture has lot of backgrounds making it extremely difficult to see the expression of F protein in cells. A much clearer picture/data is expected.

【Response】I reexamined and Figure 4 was changed.

Reviewer 2 Report

The manuscript by Dr. Nakayama Group described a new chimeric MV/RSV vaccine candidate, where ectodomain of F and HA proteins were replaced by RSV's. The overall research design is soundness and results are straightforward, although some major issues should be addressed before publication.
1) The author showed clearly the design of the vaccine construction, which technically should be regarded as MV-like particles. However, then the main issue is that if the modified MV-RSV chimeric can form the stable MV-like particles, and the evidence for the display of antigens (RSV-HA/F) is also required. Thus, TEM and immuno-TEM using HA/F antibody are suggested.
2) All figures are presented in low quality. It is better to re-organized and adjust the ratio/size.
3) Scale bar is required in Figs. 3, 4, 5, and 7.
4) It seems that this MS was prepared in a hurry and there are lots of typos and thus English editing and formating are needed.

Author Response

Reviewer 2: Thank you for your comments

【Comment】1) The author showed clearly the design of the vaccine construction, which technically should be regarded as MV-like particles. However, then the main issue is that if the modified MV-RSV chimeric can form the stable MV-like particles, and the evidence for the display of antigens (RSV-HA/F) is also required. Thus, TEM and immuno-TEM using HA/F antibody are suggested.

【Response】We could not perform electron microscopic examination. From the results of infection experiment, we suppose chimeric MV/RSV forms infectious virus particles.

【Comment】2) All figures are presented in low quality. It is better to re-organized and adjust the ratio/size.
【Response】 All figures were revised with higher resolution. Figure 4 was re-examined and changed new figure.

【Comment】3) Scale bar is required in Figs. 3, 4, 5, and 7.

【Response】 Scale bars were not applicable and magnification was provided.

【Comment】4) It seems that this MS was prepared in a hurry and there are lots of typos and thus English editing and formatting are needed.

【Response】I carefully revised.

Reviewer 3 Report

In their paper, Sawada et al expand their previous studies on chimeric AIK-C/RSV vaccines, dating back to from 2011 onwards. This time, they replaced F and HA proteins in AIK-C by F and G proteins from RSV and charactertised the new construct in this paper. They prove that the chimera works through several non-lymphoid cell lines. Furthermore, the intramuscular administration of the chimera significantly decreases the titre of the virus upon subsequent infection with RSV of measles.

I have to say, they way things are written confuses me a bit. Throughout the paper, there is very little distinction between what has been done on the previous studies and the current one. The other moment is the experimental design. The Authors seem to have three groups of samples (MVAIK, MV/RSV, RSV/Long) in one sets of experiments and only two groups in other sets. This needs balancing. Polishing the writing style would not harm either.

My suggestions:

1. Title: needs to be shorter

2. Abstract: Needs clarifying what is related to the previous study, what is new. I would suggest not to put too much info on the previous work in the Abstract.

3. Intro:

I suggest to expand the last sentence to detail what sides of the immune response were looked at in the paper.

4. Methods:

Why were the antibiotics present in the Vero cell medium? Please add the microscopy details.

Results:

5. Please clearly define MVAIK, MV/RSV, RSV/Long.

6. Does RSV/Long survive 40+C conditions?

7. Fig3: please add scale bars. Same comment applies to other figures with microscopic images

8. Fig4: can't se any fluorescence. Is there MVAIK data?

9. Fig6: Are all points in the four datasets equal to zero? If not, could they be plotted e.g. Using a second Y axis?

10. Fig7: Is there a way to plot  MVAIK alongside MV/RSV, for comparison. Histological data: what features are we looking at (arrows)?

Author Response

Reviewer 3: Thank you for your comments

Throughout the paper, there is very little distinction between what has been done on the previous studies and the current one. The other moment is the experimental design. The Authors seem to have three groups of samples (MVAIK, MV/RSV, RSV/Long) in one sets of experiments and only two groups in other sets. This needs balancing. Polishing the writing style would not harm either.

My suggestions:

【Comment】1. Title: needs to be shorter

【Response】Title was changed as following: Chimeric measles virus (MV/RSV), having ectodomains of respiratory syncytial virus (RSV) F and G proteins instead of measles envelop proteins, induced protective antibodies against RSV

【Comment】2. Abstract: Needs clarifying what is related to the previous study, what is new. I would suggest not to put too much info on the previous work in the Abstract.

【Response】Some parts of the previous study were deleted and abstract was rewritten as following;

In our previous study, fusion (F) or glyco (G) protein coding sequence of respiratory syncytial virus (RSV) was inserted at the P/M junction of the measles AIK-C vector (MVAIK), and the recombinant measles virus induced protective immune responses. In the present study, the ectodomains of measles fusion (F) and hemagglutinin (HA) proteins were replaced with those of RSV F and G proteins, and a chimeric MV/RSV vaccine was developed. It expressed F and G proteins of RSV and induced cytopathic effect (CPE) in epithelial cell lines (Vero, A549, and HEp-2 cells) but not in lymphoid cell lines (B95a, Jurkat, and U937 cells). A chimeric MV/RSV grew similarly to AIK-C, with no virus growth at 39ºC. It induced NT antibodies against RSV in cotton rats 3 weeks after immunization through intramuscular route, and enhanced response was observed after the second dose at 8 weeks after the first dose. After the RSV challenge with 106 PFU, significantly lower virus (101.4±0.1 PFU of RSV) was recovered from lung tissue in the chimeric MV/RSV vaccine group than in the MVAIK control group with 104.6±0.2 PFU, and no obvious inflammatory pathological finding was noted. The strategy of ectodomain replacement in measles virus vector is expected to lead to the development of safe and effective vaccines for other enveloped viruses.

【Comment】3. Intro: I suggest to expand the last sentence to detail what sides of the immune response were looked at in the paper.

【Response】This paragraph was revised as following: Measles virus is classified in subfamily Paramyxovirinae, and has two functional envelop proteins, F and HA proteins. RSV and measles virus belong to the same family Paramyxoviridae and have a similar genome structure of F and G/HA genomes. Conventional approach was failed to develop RSV vaccine candidate having temperature sensitivity (ts) phenotype. AIK-C measles vaccine has unique ts characteristics, contributing to the attenuation mechanisms and MVAIK constructed by reverse genetics also has ts phenotype [22]. MVAIK/RS/F or MVAIK/RS/G has measles envelop proteins of F and HA and the efficacy may be influence by the pre-existing antibodies. In the present study, chimeric RSV vaccine candidate was constructed based on the AIK-C measles virus, replacing the ectodomains of the F and HA proteins of measles virus with those of the F and G proteins of RSV. The chimeric measles and RSV vaccine (MV/RSV) was characterized, and we investigated the immune response.

【Comment】4. Methods: Why were the antibiotics present in the Vero cell medium? Please add the microscopy details.

【Response】 All cell lines were maintained adding antibiotics to prevent bacterial contamination. Microscopic images were taken using EVOS FL light microscope (Life technologies, USA).

【Comment】5. Results: Please clearly define MVAIK, MV/RSV, RSV/Long.

【Response】Line 104-106, MVAIK was defined: Backbone of MVAIK was constructed from AIK-C vaccine strain, developed in our laboratory, which has been used as recommend immunization in Japan since 1977, having ts phenotype [22].

Line 84-86, chimeric RSV vaccine candidate was constructed based on the AIK-C measles virus, replacing the ectodomains of the F and HA proteins of measles virus with those of the F and G proteins of RSV.

RSV/Long is a laboratory reference for RSV subgroup A.

【Comment】6. Does RSV/Long survive 40ºC conditions?

【Response】RSV Long strain grows well at 39ºC. I suppose this might be related to the failure in development of vaccine candidate having ts phenotype.

【Comment】7. Fig3: please add scale bars. Same comment applies to other figures with microscopic images

【Response】 Scale bars were not applicable and magnification was provided.

【Comment】8. Fig4: cannot see any fluorescence. Is there MVAIK data?

【Response】I reexamined and Figure 4 was changed. From the results in Figure 3, MV/AIK did not express RSV F protein and it was not examined.

【Comment】9. Fig6: Are all points in the four datasets equal to zero? If not, could they be plotted e.g. Using a second Y axis?

【Response】Y axis 0 means no detectable antibody.

【Comment】10. Fig7: Is there a way to plot MVAIK alongside MV/RSV, for comparison. Histological data: what features are we looking at (arrows)?

【Response】Figure 7 was changed.

Round 2

Reviewer 1 Report

The revised version of the MS slightly improved but does not meet the publication quality.  

I believe that the statistical analysis are required for some of their experiments and Figure 4 is still not clear to me. 

First weakness I would point out, is the absence of statistical analysis in this study, however in the abstract the authors mention that " After the RSV challenge with 106 PFU, significantly lower virus (101.4± 31 0.1 PFU of RSV) was recovered from lung 32 tissue in the chimeric MV/RSV vaccine group than in the MVAIK control group with 104.6± 33 0.2 PFU, and no obvious inflammatory pathological finding was noted". Therefore, I still, don't understand how we can determine any experimental data to be significantly lower or higher without performing any statistical analysis. Another weakness is figure 4 where authors have shown the live infected Hep-2 cells staining to determine the expression of F protein by using monoclonal antibody against RSV F protein coupled with secondary antibody against mouse IgG conjugated with FITC. Still this picture is not clear in the revised manuscript. Authors needs to use appropriate positive and negative controls as background noise are extremely high. Authors also needs to elaborate the descriptions of all the figures including information like how those were performed and how many times repeats conducted etc.   

Reviewer 2 Report

I raised an issue that it is questionable to use a modified viral construct as a vaccine candidate without verification of the morphology of the resulting product. I understand the antibody production is essential to criticize the vaccine candidate. Without clear evidence of viral structure, either in presence of viral morphology or infection efficiency, it is difficult to draw a conclusion. The author argued that the result of immuno-staining should explain the ability of the viral infection. However, the quality of Figure 4 is dim and thus not sufficient to convince me. I suggested TEM to confirm the morphology in the last round of revision. If they have trouble in doing so, I think the sucrose gradient centrifugation method [control (WT), chimeric] combining with western blot is also an option for authors. I hope the authors can address this issue carefully to improve their manuscript.

Reviewer 3 Report

I think the Authors have dealt adequately with the queries raised, the paper should be accepted now.